# Expression of β-Glucosidases from the Yak Rumen in Lactic Acid Bacteria: A Genetic Engineering Approach

**DOI:** 10.3390/microorganisms11061387

**Published:** 2023-05-25

**Authors:** Chuan Wang, Yuze Yang, Chunjuan Ma, Yongjie Sunkang, Shaoqing Tang, Zhao Zhang, Xuerui Wan, Yaqin Wei

**Affiliations:** 1College of Veterinary Medicine, Gansu Agricultural University, Lanzhou 730070, China; wangchuan@gsau.edu.cn (C.W.);; 2Center for Anaerobic Microbes, Institute of Biology, Gansu Academy of Sciences, Lanzhou 730000, China; 3Beijing Animal Husbandry Station, Beijing 100107, China

**Keywords:** β-glucosidase, fusion expression, *L. lactis* NZ9000, enzyme activity

## Abstract

β-glucosidase derived from microorganisms has wide industrial applications. In order to generate genetically engineered bacteria with high-efficiency β-glucosidase, in this study two subunits (*bglA* and *bglB*) of β-glucosidase obtained from the yak rumen were expressed as independent proteins and fused proteins in lactic acid bacteria (*Lactobacillus lactis* NZ9000). The engineered strains *L. lactis* NZ9000/pMG36e-usp45-*bglA*, *L. lactis* NZ9000/pMG36e-usp45-*bglB*, *and L. lactis* NZ9000/pMG36e-usp45-*bglA*-usp45-*bglB* were successfully constructed. These bacteria showed the secretory expression of BglA, BglB, and Bgl, respectively. The molecular weights of BglA, BglB, and Bgl were about 55 kDa, 55 kDa, and 75 kDa, respectively. The enzyme activity of Bgl was significantly higher (*p* < 0.05) than that of BglA and BglB for substrates such as regenerated amorphous cellulose (RAC), sodium carboxymethyl cellulose (CMC-Na), desiccated cotton, microcrystalline cellulose, filter paper, and 1% salicin. Moreover, 1% salicin appeared to be the most suitable substrate for these three recombinant proteins. The optimum reaction temperatures and pH values for these three recombinant enzymes were 50 °C and 7.0, respectively. In subsequent studies using 1% salicin as the substrate, the enzymatic activities of BglA, BglB, and Bgl were found to be 2.09 U/mL, 2.36 U/mL, and 9.4 U/mL, respectively. The enzyme kinetic parameters (*V*max, *K*m, *K*cat, and *K*cat/*K*m) of the three recombinant strains were analyzed using 1% salicin as the substrate at 50 °C and pH 7.0, respectively. Under conditions of increased K^+^ and Fe^2+^ concentrations, the Bgl enzyme activity was significantly higher (*p* < 0.05) than the BglA and BglB enzyme activity. However, under conditions of increased Zn^2+^, Hg^2+^, and Tween20 concentrations, the Bgl enzyme activity was significantly lower (*p* < 0.05) than the BglA and BglB enzyme activity. Overall, the engineered lactic acid bacteria strains generated in this study could efficiently hydrolyze cellulose, laying the foundation for the industrial application of β-glucosidase.

## 1. Introduction

β-glucosidase (EC3.2.1.21) is a cellulase that has been discovered in plants, animals, and microorganisms [1]. It is a dual-function enzyme that hydrolyzes glycosidic bonds in alkyl-, amino-, and aryl-β-D-glucosides, as well as in cyanogenic glucosides. In addition, β-glucosidase can also catalyze the formation and degradation of glycosyl bonds between distinct compounds via transglycosylation [2,3]. Unsurprisingly, β-glucosidase had important applications in the food and medicine industries, as well as in energy and feed processing [4,5,6]. β-glucosidase has been isolated from *Xylella rickettsii* [7], *Thermomucor indicae-seudaticae* [8], and *Acidothermus cellulolyticus* [9]. Moreover, the β-glucosidase gene has been isolated from a metagenomic library of cattle rumen [10].

The structure and enzymatic properties of β-glucosidase vary widely depending on its source [11]. Although bacteria and fungi have been used for industrial β-glucosidase production, the β-glucosidase from natural microbial strains has disadvantages such as a low yield, low enzyme activity, and poor thermal stability [12,13].

The overexpression and purification of recombinant proteins via genetic engineering have enabled the widespread application of these proteins [14]. Heterologous expression systems have been adopted to increase β-glucosidase activity. Different β-glucosidase genes have been heterologously expressed in microbial systems such as *Lactobacillus lactis* [14], *Escherichia coli* [15], and *Pichia pastoris* [16]. In particular, *L. lactis*, which is generally regarded as a safe (GRAS) microorganism and has probiotic properties, is used to express heterologous genes [17,18]. The pMG36e plasmid has been derived from pVW01 and replicated in *Lactobacillus* spp., *Lactococcus* spp., *Bacillus subtilis*, and *E. coli* [19]. In addition, in order to obtain a cellulase consortium containing different cellulolytic activities, fusion expression has been applied for the synthesis of bifunctional enzymes that enable cellulose hydrolysis [20,21]. It has been proven that individually expressed, fused, and co-expressed endoglucanases and β-glucosidases can promote the hydrolysis of sugarcane bagasse [22].

The yak rumen is inhabited by unique, complex, diverse, and large microbial communities that co-metabolize and efficiently degrade wild grass to provide both energy and nutrients for yaks. Hence, the yak rumen is rich in highly active lignocellulose hydrolases [23,24]. Many studies showed that microbial communities in the yak rumen are one of the main sources of cellulase [25,26,27]. In order to generate genetically engineered bacteria producing high-efficiency β-glucosidase, in this study we cloned two β-glucosidase genes *bglA* and *bglB* from the bovine rumen into the constitutive expression vector pMG36e. These genes were cloned to encode independent proteins and fused proteins; accordingly, the expression vectors pMG36e-usp45-*bglA*, pMG36e-usp45-*bglB*, and pMG36e-usp45-*bglA*-usp45-*bglB* were constructed. These vectors were inserted into the host strain *L. lactis* NZ9000 to develop three bacterial expression systems. Subsequently, the hydrolysis activity of the β-galactosidases and the optimal growth conditions of the three recombinant strains were analyzed.

## 2. Materials and Methods

### 2.1. Bacterial Strains, Plasmids, and Culture Conditions

The bacterial strains and plasmids used in this study are listed in Table 1. *Escherichia coli* DH5α was cultured in Luria-Bertani (LB) medium (both agar and broth) at 37 °C. Meanwhile, *L. lactis* NZ9000 was cultured in GM17 medium at 30 °C. Then, 0.1 μg/mL of erythromycin (Takara Biotechnology Co., Ltd., Dalian, China) was used to screen for recombinant *L. lactis* NZ9000 containing the pMG36e vector.

### 2.2. Construction of Individual Genes and the Fusion Gene

Primers were designed based on the NCBI *bglA* and *bglB* sequences (GenBank Accession No. M60210.1 and M60211.1) (Table 2). Using the CTAB method, the total genomic DNA was extracted from the yak rumen fluid collected from 20 Tianzhu yaks aged 5–6 years (gibbed and male, body weight of about 250 kg, grazed on wild grass at the Wushaoling pasture in the Tibetan Autonomous County of Tianzhu, Gansu Province, China). Both *bglA* and *bglB* were amplified via polymerase chain reaction (PCR) with the bglA-F/R and bglB-F/R primers. PCR was performed with a total volume of 50 μL, containing 2 μL of EasyPfu Buffer Polymerase (TransGen Biotech Co., Ltd. Beijing, China), 5 μL 10× EasyPfu Buffer, 5 μL 2.5 mmol/L dNTPs, 2 μL ligation product, 26 μL nuclease-free water, and 2.5 μL of each primer. The PCR procedure was as follows: pre-denaturation at 95 °C for 5 min; 35 cycles of 95 °C for 30 s, 55 °C for 30 s, and 72 °C for 1 min 40 s; and final extension at 72 °C for 10 min. The PCR products of *bglA* and *bglB* were purified using the Easy Pure PCR Purification Kit (TransGen Biotech Co., Ltd., Beijing, China).

The primers of bglA-R1 and bglB-F1 were phosphorylated using the T4 polynucleotide kinase enzyme (Beyotime Institute of Biotechnology, Jiangsu, China). Then, the *bglA* and *bglB* fragments were amplified by PCR using the *bglA*-F/R1 and *bglB*-F1/R primers and the total genomic DNA as a template. The PCR system and procedure were the same as those described above. The PCR products of *bglA* and *bglB* were ligated via T4-DNA ligase (TakaRa Biotechnology Co., Ltd. Dalian, China). The fusion *bglA-bglB* fragment was amplified using bglA-R1 and bglB-F1 specific primers with the ligation product as a template. PCR was performed with a total volume of 50 μL, containing 2 μL of EasyPfu Buffer Polymerase (TransGen Biotech Co., Ltd. Beijing, China), 5 μL of 10× EasyPfu Buffer, 5 μL of 2.5 mmol/L dNTPs, 2 μL of ligation product, 26 μL of nuclease-free water, and 2.5 μL of each primer. The PCR procedure was as follows: pre-denaturation at 95 °C for 5 min; 35 cycles of 95 °C for 30 s, 58 °C for 30 s, and 72 °C for 3 min; and final extension at 72 °C for 10 min. Then, *bglA-bglB* was recovered using the Easy Pure Quick Gel Extraction Kit (TransGen Biotech Co., Ltd. Beijing, China).

### 2.3. Construction and Transformation of the Recombinant Plasmid

The PCR products of purified *bglA*, *bglB*, and *bglA-bglB* were digested with *Pst* I and *Hind* III (TakaRa Biotechnology Co., Ltd. Dalian, China). The same enzymes were used to digest pMG36e and the products were linked to the vector using T4-DNA ligase (TakaRa Biotechnology Co., Ltd. Dalian, China) to produce the recombinant vectors pMG36e-usp45-*bglA*, pMG36e-usp45-*bglB*, *and* pMG36e-usp45-*bglA*-usp45-*bglB.* Then, the recombinant vectors were introduced into the *E. coli* DH5α cells. Positive *E. coli* DH5α/pMG36e-usp45-*bglA*, *E. coli* DH5α/pMG36e-usp45-*bglB*, and *E. coli* DH5α/pMG36e-usp45-*bglA*-usp45-*bglB* transformants were verified using PCR and double enzyme digestion (*Pst* I and *Hind* III). The verified and correct recombinant plasmids were selected for sequencing and the sequencing results were compared with data from NCBI’s BLAST tool.

Using the Gene Pulser Xcell (Bio-Rad Laboratories, Inc, Hercules, CA, USA), the verified recombinant plasmids pMG36e-usp45-*bglA*, pMG36e-usp45-*bglB*, *and* pMG36e-usp45-*bglA*-usp45-*bglB* were inserted into *L. lactis* NZ9000 via electroporation under the following conditions: a pulse voltage of 2200 V, pulse resistance of 100 Ω, pulse capacitance of 25 μF, and pulse duration of 3 s. Positive transformants were screened using 0.1 μg/mL of erythromycin selection and verified by PCR.

The inoculum of *L. lactis* NZ9000/pMG36e-usp45-*bglA*, *L. lactis* NZ9000/pMG36e-usp45-*bglB*, and *L. lactis* NZ9000/pMG36e-usp45-*bglA*-usp45-*bglB* was added to GM17 plates containing 1 g/L of carboxymethyl cellulose (CMC) and 0.1 μg/mL of erythromycin (pH 7.0). The plates were cultured at 30 °C for 48 h. The plates were then treated with 1 g/L of Congo red solution. After incubation for 30 min, the plates were washed with 1 mol/L NaCl to reveal clear zones against a red background. The clear zones were indicative of CMC hydrolysis.

### 2.4. Enzyme Assays and Protein Analysis of Recombinant L. lactis NZ900

The recombinant strains were incubated at 30 °C for 36 h, and, then, the supernatant was collected. The recombinant proteins were purified using the precipitation method [29]. The molecular masses of enzymes from the recombinant strains were estimated via sodium dodecyl sulfate–polyacrylamide gel electrophoresis (SDS-PAGE). With glucose as a standard, the activity of all of these enzymes was determined by measuring the amount of reducing sugars released after enzyme catalysis using the 3,5-dinitrosalicylic acid (DNS) method [30]. Regenerated amorphous cellulose (RAC), sodium carboxymethyl cellulose (CMC-Na), dried cotton, microcrystalline cellulose, filter paper [31], and 1% salicin were used as substrates to test the cellulase activity of individual and recombinant fusion enzymes. Recombinant proteins inactivated by boiling were used as blank controls.

### 2.5. Enzyme Assays at Different Temperatures and pH Values

The optimum substrate was reacted with the supernatant of the recombinant enzyme at different temperatures (30–90 °C) for 30 min to determine the optimum reaction temperature for the recombinant enzyme. The optimum substrate was reacted with the recombinant enzyme supernatant in several different pH buffers (pH 3.0–9.0) at the optimum temperature for 30 min to determine the optimum pH for recombinase activity.

### 2.6. Enzyme Kinetic Parameter Assays

The activity of recombinant enzymes was measured using various concentrations (0.01–10 nmol/L) of salicin as the substrate at 50 °C and pH 7.0. The enzyme kinetic parameters (*V*max, *K*m, *K*cat, and *K*cat/*K*m) were estimated using GraphPad Prism 8.4.2 with Michaelis–Menten equation fitting and Lineweaver–Burk plots (double-reciprocal plots).

### 2.7. Enzyme Assays at Different Metal Ion Concentrations

Furthermore, 1% salicin was used as the substrate at 50 °C and pH 7.0, K^+^, Mg^2+^, Mn^2+^, Zn^2+^, Hg^2+^, Fe^2+^, Co^2+^, Ca^2+^, and Cu^2+^ were added to the recombinant enzyme-containing supernatant and the optimal substrate such that the final concentration of the solutions was 1 mmol/mL or 5 mmol/mL. Tween 20 and Tween 80 solutions were added to the recombinant enzyme-containing supernatant and optimal substrate such that the final concentration of the solutions was 1% or 10%. These solutions were incubated at the optimal temperature and pH for 30 min. The effect of different concentrations of ions and solutions on the activity of the recombinant enzyme was measured. Solutions without metal ions and chemical reagents were used as blank controls.

### 2.8. Statistical Analysis

To evaluate the influence of various temperatures, pH values, and ions and chemical reagents on the enzymatic activity of the recombinant enzymes, the data were statistically analyzed using GraphPad Prism 8.4.2 one-way ANOVA. Enzyme assays were performed in triplicates. The means were compared using Tukey’s test. The significance was set at *p* < 0.05.

## 3. Results

### 3.1. Identification of Recombinant Plasmids

Maps of the three recombinant plasmids generated in this study using SnapGene software (Version 5.2.4) are shown in Figure 1a–c. The recombinant plasmid contained the promoter P32, the signal peptide usp45, and a ribosomal binding site (RBS). pMG36e-usp45-*bglA*, pMG36e-usp45-*bglB*, and pMG36e-usp45-*bglA*-usp45-*bglB* were successfully constructed and verified via double digestion (*Pst* I and *Hind* III). The sequencing showed that the size of *bglA*, *bglB*, and fusion *bglA-bglB* was 1361 bp, 1361 bp, and 2722 bp, respectively (Figure 1d). Using Blast on NCBI, it was found that the *bglA* and *bglB* sequences are 100% identical to *Paenibacillus polymyxa* (M60210.1 and M60211.1).

As shown in Figure 2, *L. lactis* NZ9000/pMG36e-usp45-*bglA*, *L. lactis* NZ9000/pMG36e-usp45-*bglB*, and *L. lactis* NZ9000/pMG36e-usp45-*bglA*-usp45-*bglB* induced clear circles around them after Congo red staining. These circles indicated that substrate hydrolysis had occurred. The findings confirmed the successful construction of the three recombinant lactic acid bacteria. Notably, the hydrolysis circle of *L. lactis* NZ9000/pMG36e-usp45-*bglA*-usp45-*bglB* was larger than that of *L. lactis* NZ9000/pMG36e-usp45-*bglA* and *L. lactis* NZ9000/pMG36e-usp45-*bglB*.

### 3.2. Enzyme Expression Levels in Recombinants

*L. lactis* NZ9000/pMG36e-usp45-bglA-usp45-bglB, *L. lactis* NZ9000/pMG36e-usp45-bglA, and *L. lactis* NZ9000/pMG36e-usp45-bglB showed the secretory expression of the recombinant proteins Bgl, BglA, and BglB, respectively, as determined using SDS-PAGE. The results of the analysis using Quantity One 1-D software (version 4.6.2) showed that the band with a molecular mass of 75 kDa was Bgl. Moreover, BglA and BglB were found to have a molecular mass of 55 kDa (Figure 3).

The enzymatic activity of Bgl was significantly higher than that of BglA and BglB for various substrates. However, the difference in enzymatic activity between BglA and BglB was not significant. Among the substrates tested, the recombinant enzyme had the highest specificity for 1% salicin, followed by CMC-Na. The lowest specificity was observed for absorbable cotton. Using 1% salicin as the substrate, the enzyme activities of BglA, BglB, and Bgl in the supernatant were found to be 2.09 U/mL, 2.36 U/mL, and 9.4 U/mL, respectively. The total enzyme activities of BglA, BglB, and Bgl for cellulose were found to be 0.97 U/mL, 1.12 U/mL, and 2.53 U/mL, respectively, using filter paper as the substrate (Figure 4a).

### 3.3. Determination of Optimum Reaction Temperature and pH of the Recombinant Enzyme

The enzyme activity of Bgl, BglA, and BglB increased with an increase in temperature within the reaction temperature range of 30–50 °C. The highest enzymatic of Bgl, BglA, and BglB was observed at 50 °C. As the temperature increased beyond 50 °C, the enzyme activity showed a slow decreasing trend. Therefore, the optimum reaction temperature for the fusion enzyme was determined to be 50 °C. The recombinant enzyme activity was around 75% when the temperature was 45–65 °C (Figure 4b). The optimum reaction temperature for these 3 recombinant enzymes was 50 °C.

When the pH was 3.0–7.0, the enzyme activity increased with an increase in pH. The optimal pH for Bgl, BglA, and BglB ranged from 6.0 to 8.0. The activity of these enzymes was above 90% within this pH range and peaked at pH 7.0. When the pH was over 7.0, the enzyme activity decreased, indicating that these enzymes were neutral enzymes (Figure 4c). The optimum reaction pH for these three recombinant enzymes was 7.0.

### 3.4. Effect of Various Substrates on the Kinetic Parameters of Recombinant Enzymes

We used salicin as the substrate to determine the kinetic parameters of the recombinant enzymes at an optimal temperature and pH (Table 3). The *V*max values were 1343, 1254, and1347 μmol∙min^−1^mg^−1^ for the bacterial supernatant of *L. lactis NZ9000*/pMG36e-usp45-*bglA*, *L. lactis NZ9000*/pMG36e-usp45-*bglB*, and *L. lactis NZ9000*/pMG36e-usp45-*bglA*-usp45-*bglB*, respectively. The *K*m values were 38.61, 43.99, and 34.8 μmol∙L^−1^ for the bacterial supernatant of *L. lactis NZ9000*/pMG36e-usp45-*bglA*, *L. lactis NZ9000*/pMG36e-usp45-*bglB*, and *L. lactis NZ9000*/pMG36e-usp45-*bglA*-usp45-*bglB*, respectively. The *K*cat values were 434.8, 348.7, and 1016 s^−1^ for the bacterial supernatant of *L. lactis NZ9000*/pMG36e-usp45-*bglA*, *L. lactis NZ9000*/pMG36e-usp45-*bglB*, and *L. lactis NZ9000*/pMG36e-usp45-*bglA*-usp45-*bglB*, respectively. The *K*cat/*K*m values were 11.26, 7.93, and 29.2 L∙s^−1^∙mol^−1^ for the bacterial supernatant of *L. lactis NZ9000*/pMG36e-usp45-*bglA*, *L. lactis NZ9000*/pMG36e-usp45-*bglB*, and *L. lactis NZ9000*/pMG36e-usp45-*bglA*-usp45-*bglB*, respectively.

### 3.5. Effect of Ions and Chemical Reagents on Recombinant Enzymes

The increase in K^+^ and Fe^2+^ ion concentrations significantly increased the activity of the fusion recombinant enzyme but not the single recombinant enzymes. The activity of the Bgl enzyme was much greater than that of the BglA and BglB enzymes. Cu^2+^, Co^2+^, Mn^2+^, and Tween 80 did not affect the activity of the three recombinant enzymes. Ca^2+^ did not affect the activities of the three recombinases at a final concentration of 1 mmol/L but significantly promoted enzyme activity at a final concentration of 5 mmol/L. The recombinant enzyme was inhibited by Zn^2+^, Hg^2+^, and Tween 20, and an increase in their concentrations inhibited Bgl more significantly than BglA and BglB (Figure 5a,b).

## 4. Discussion

Currently, β-glucosidase is widely used in industrial applications, including in the textile, food processing, biorefinery, brewing, chemical, detergent, animal feed, paper, pharmaceutical, medical, pulp, and waste management industries [32,33]. Microbial communities in the yak rumen are one of the main sources of cellulase. For example, *Enterococcus faecalis* EF85 and EF83 isolated from the rumen of Tibetan yaks have the potential to degrade cellulose [25]. In addition, high-throughput 16S rRNA sequencing and metagenomic analysis have indicated that the rumen microbiome of plateau yaks has several genes encoding cellulases and hemicelluloses [26,27]. In order to obtain high-efficiency β-glucosidase, in this study two β-glucosidase genes *bglA* and *bglB* from the yak rumen were cloned into pMG36e as independent or fused proteins to construct the expression vectors pMG36e-bglA, pMG36e-bglB, and pMG36e-bglA-bglB. The engineered strains *L. lactis* NZ9000/pMG36e-usp45-*bglA*, *L. lactis* NZ9000/pMG36e-usp45-*bglB*, and *L. lactis* NZ9000/pMG36e-usp45-*bglA*-usp45-*bglB* were successfully constructed thereafter and showed the secretory expression of BglA, BglB, and Bgl, respectively. pMG36e was constructed based on the transcriptional and translational signals of the *L. lactis* milk lipid subspecies protease gene. It contained a promoter, an RBS, the start of an open reading frame, and a transcriptional terminator, and it was known to be a constitutive expression vector for the inserted gene in *L. lactis* [19]. pMG36e has previously been used to successfully express β-galactosidase in *L. lactis* [34]. *L. lactis* NZ9000 was a manually constructed recombinant host with the nisin-controlled gene expression (NICE) system [28]. In previous studies, *L. lactis* NZ9000 was used to express various proteins originating from other bacteria [35]. *L. lactis*, which is a well-recognized probiotic, has advantages over other expression systems in expressing recombinant proteins [28,36]. Unlike *E. coli*, *L. lactis* has a single outer membrane and can secrete target proteins directly into the extracellular environment [37]. The total enzyme production in *L. lactis* is higher than that in *E. coli* [38]. The ability of *L. lactis* to efficiently express and secrete proteins in large quantities is dependent on the selection of suitable and effective transcriptional promoters and secretory signal peptides [39]. Usp45, the main signal peptide for the extracellular proteins of *L. lactis*, plays an important role in guiding proteins to the cytoplasmic membrane [40]. By replacing the natural Nuc signal peptide with Usp45, the efficiency of heterologous protein secretion by *L. lactis* can be increased greatly [41].

The biodegradation of cellulose requires multiple cellulases that act in concert with each other to make cellulose degradation more efficient [42]. The β-glucosidase of *Bacillus subtilis* SU40 acts in synergy with the endoglucanase of *Bacillus subtilis* MG7 to completely degrade rice straw [41]. Fusion proteins containing xylanases and endonucleases with multiple catalytic structural domains can improve the efficiency of cellulose degradation [43]. In one study, three different cellulase genes were fused to construct a BCE with trifunctional cellulase activity. The fused enzyme showed higher specific activities of the β-glucosidase BG, endoglucanase CBH, and exoglucanase EG than did single enzymes under the same conditions [22]. In the present study, the enzymatic activity of the recombinant fusion enzyme in this study was 9.4 U/mL, which was approximately 3.5-fold higher than that of a single cellulase in lactic acid bacteria.

The temperature optima of cellulases in currently available commercial enzyme cocktails is typically around 50 °C [44]. In this study, an analysis of the enzymatic properties of the recombinant cellulase showed that the optimum reaction pH was 7.0 and the optimum reaction temperature was 50 °C. The highest growth and maximum CMCase production by *Bacillus licheniformis* TLW-3 were recorded at pH 7 and 50 ºC [45]. The Bgl enzyme in this study had a longer activity duration than the bacterial β-glucosidase of *Aeromonas alternata*, which rapidly lost its activity after incubation at 40 °C [46]. The Bgl in this study also exhibited more pH tolerance than the β-glucosidase *unglu135B12*, which showed optimal enzymatic activity at pH 5.0 [10]. In the present study, cellulase activity increased when the temperature increased from 30 °C to 50 °C and decreased when the temperature was above 80 °C. This was because as the temperature increased, the effective collision rate between the substrate molecules and the enzyme became greater, and the reaction rate became faster. However, when the temperature was too high, the denaturation of the enzyme was accelerated, leading to a decrease in enzyme activity [47]. The effect of metal ions on cellulase activity was specific to the cellulase being examined.

Different metal ions may have different affinities for different amino acid residues in cellulose, leading to conformational changes that may stimulate or inhibit cellulase activity. In this study, K^+^ and Fe^2+^ promoted the activity of the recombinant enzyme, while Cu^2+^, Co^2+^, and Mn^2+^ had little effect on this activity. Meanwhile, Hg^2+^ and Zn^2+^ inhibited the activity of the recombinant enzyme. Hg^2+^ is a heavy metal ion that can disrupt the structure of enzymes and even inactivate them [48]. Fe^2+^, Co^2+^, and Mn^2+^ can stimulate the enzymatic activity of endoglucanase and xylanase isolated from the rumen of goats [49]. The activity of β-glucosidase isolated from *Mycobacterium thermophilum* was found to be strongly reduced in the presence of Hg^2+^ and Cu^2+^ ions [50]. Interestingly, the fused Bgl examined in this study had higher enzymatic activities that the single cellulases. This was probably because the fused cellulase disrupted the crystalline structure of cellulose more deeply, increasing accessibility and making the accessible surface area larger. When the pores of the substrate were large enough, the fused enzyme enhanced its adsorption of cellulose, which improved the hydrolysis efficiency of the cellulase [51]. In a study on cellulose-secreting engineered *L. lactis* that promote lignocellulose degradation in high-moisture alfalfa, a combination of fusion *L. lactis* was found to be superior to a combination of cellulase and wild-type lactic acid bacteria subspecies in terms of silage quality and degradation of lignin [52].

## 5. Conclusions

In this study, the strains *L. lactis* NZ9000/pMG36e-usp45-*bglA*, *L. lactis* NZ9000/pMG36e-usp45-*bglB*, *and L. lactis* NZ9000/pMG36e-usp45-*bglA*-usp45-*bglB* were successfully engineered. They showed the secretory expression of BglA, BglB, and Bgl, respectively. The molecular weights of BglA, BglB, and Bgl were about 55 kDa, 55 kDa, and 75 kDa, respectively. In subsequent studies, using 1% salicin as the substrate, the optimum reaction temperature and pH for the three recombinant enzymes were found to be 50 °C and 7.0, respectively. The *V*max, *K*m, *K*cat, and *K*cat/*K*m of the three recombinant strains were analyzed using 1% salicin as the substrate at 50 °C and pH 7.0, respectively. The Bgl enzyme activity was significantly higher than the BglB and BglA enzyme activity (*p <* 0.05). Overall, the results showed that the β-glucosidase gene for lignocellulose degradation from the rumen of Tianzhu yak grazing on the Qinghai-Tibetan Plateau has important application and development prospects in the industrial production of β-glucosidase. Further development and research are warranted.

## Figures and Tables

**Figure 1 microorganisms-11-01387-f001:**
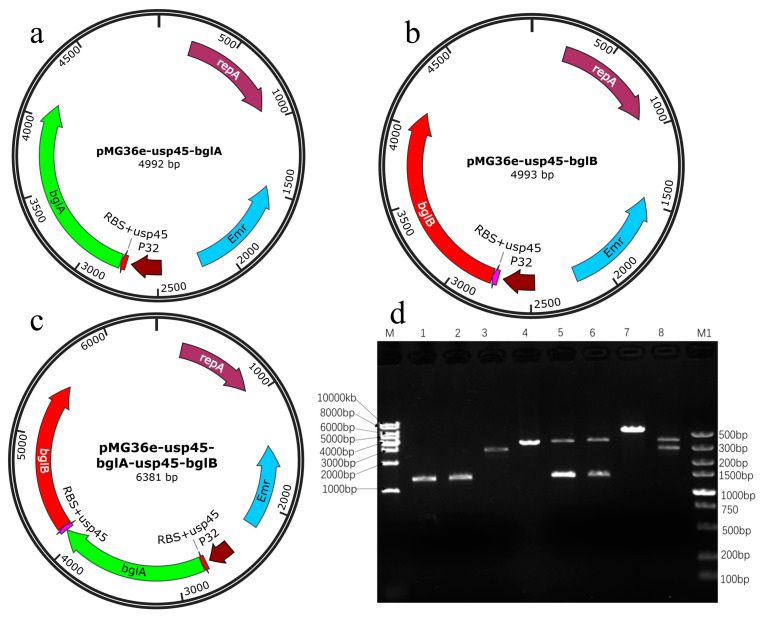
Maps of the three recombinant plasmid constructs and their electropherogram. (**a**) pMG36e-usp45-*bglA* recombinant plasmid; (**b**) pMG36e-usp45-*bglB* recombinant plasmid; (**c**) pMG36e-usp45-*bglA*-usp45-*bglB* recombinant plasmid; (**d**) M: DNA molecular weight marker; 1: PCR product of *bglA*; 2: PCR product of *bglB*; 3: PCR product of *bglA-bglB*; 4: pMG36e plasmid after *Pst* I digestion; 5: pMG36e-*bglA* plasmid after *Pst* I and *Hind* III digestion; 6: pMG36e-*bglB* plasmid after *Pst* I and *Hind* III digestion; 7: pMG36e-*bglA-bglB* plasmid after *Pst* I digestion; 8: pMG36e-*bglA-bglB* plasmid after *Pst* I and *Hind* III digestion.

**Figure 2 microorganisms-11-01387-f002:**
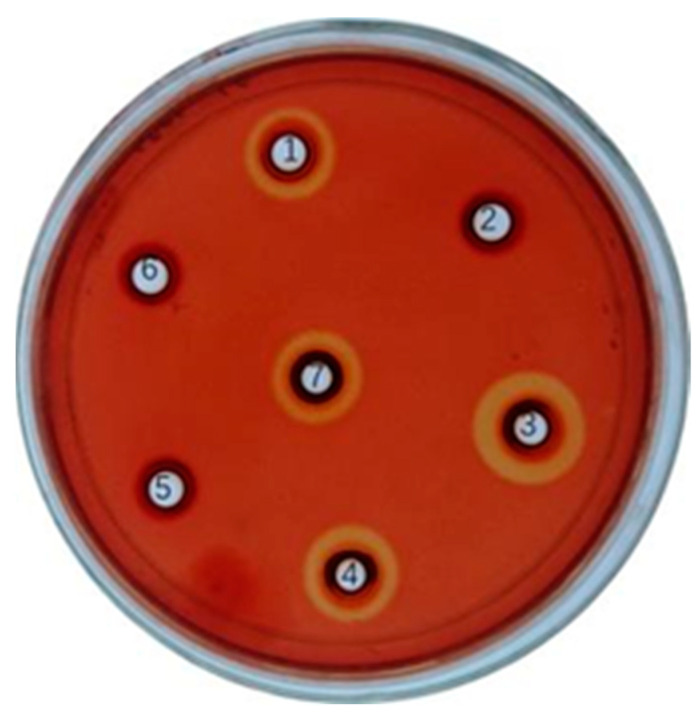
The analysis of recombinant enzymes activity using agar plates with 0.1% sodium carboxymethyl cellulose CMC-Na-stained Congo red. The halos dimeter is proportional to enzyme’s activity. To each well 20 μL of the enzyme with concentration 0.26–0.27 mg/mL of recombinant enzymes was transferred; 1: The inoculum of *L. lactis* NZ9000/pMG36e-usp45-*bglA*; 2: The inoculum of *L. lactis* NZ9000/pMG36e; 3: The inoculum of *L. lactis* NZ9000/pMG36e-usp45-*bglA*-usp45-*bglB*; 4: The inoculum of *L. lactis* NZ9000/pMG36e-usp45-*bglB*; 5: Negative control ddH_2_O; 6: The inoculum of *L. lactis* NZ9000/pMG36e; 7: β-glucosidase standard.

**Figure 3 microorganisms-11-01387-f003:**
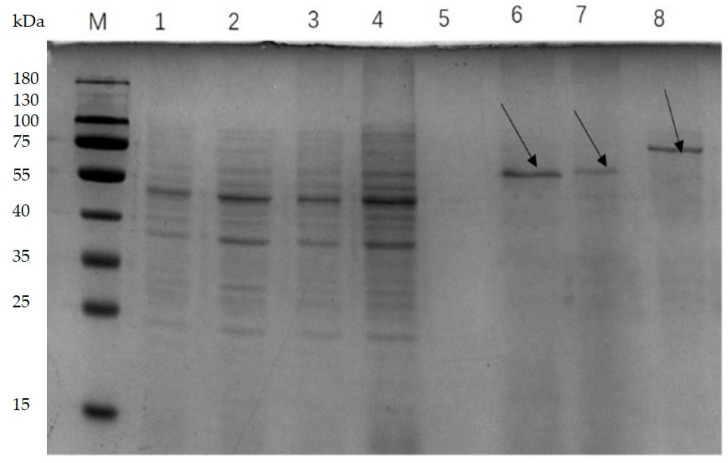
SDS-PAGE of supernatant from recombinant strains. M: Protein molecular weight marker, 1: Bacterial supernatant precipitate of *L. lactis* NZ9000/pMG36e; 2: Bacterial supernatant precipitate of *L. lactis* NZ9000/pMG36e-usp45-bglA; 3: Bacterial supernatant precipitate of *L. lactis* NZ9000/pMG36e-usp45-*bglB*; 4: Bacterial supernatant precipitate of *L. lactis* NZ9000/pMG36e-usp45-bglA-usp45-*bglB*; 5: TCA/acetone precipitated protein of pMG36e from the *L. lactis* NZ9000/pMG36e bacterial supernatant; 6: TCA/acetone-precipitated BglA protein from *L. lactis* NZ9000/pMG36e-usp45-bglA bacterial supernatant; 7: TCA/acetone-precipitated BglB protein from *L. lactis* NZ9000/pMG36e-usp45-*bglB* bacterial supernatant; 8: TCA/acetone-precipitated Bgl protein from *L. lactis* NZ9000/pMG36e-usp45-bglA-usp45-*bglB* bacterial supernatant.

**Figure 4 microorganisms-11-01387-f004:**
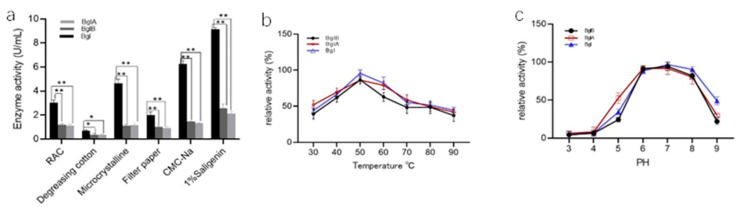
Effect of temperature, pH, and substrate on recombinant enzyme activity. (**a**) Substrate specificity of recombinant proteins; (**b**) Effect of temperature on recombinant enzyme activity; (**c**) Effect of pH on recombinant enzyme activity; (*: Represents the difference within experimental groups; *: *p* < 0.05; **: *p* < 0.01; error bars in graphs represent standard deviations).

**Figure 5 microorganisms-11-01387-f005:**
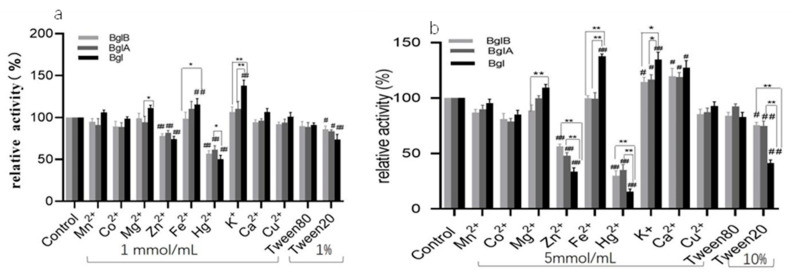
Effect of different concentrations of metal ions and chemical reagents on recombinant enzyme activity. (**a**) Effect of 1 mmol/mL ions and 1% chemical solutions on enzyme activity; (**b**) Effect of 5 mmol/mL ions and 10% chemical solutions on enzyme activity (#: Represents the difference between experimental groups and the control group; #: *p* < 0.05; ##*: p* < 0.01; ###: *p* < 0.001 *: Represents the difference within experimental groups; *: *p* < 0.05; **: *p* < 0.01; Error bars in graphs represent standard deviations).

**Table 1 microorganisms-11-01387-t001:** Bacterial strains and plasmids used in this study.

Strain and Plasmid		Relevant Trait(s)	Source or Reference
Strains	*Escherichia coli* DH5α	supE44 Δlac U169 (Φ80 lacZ ΔM15) hsdR17 recA1, endA1 gyrA96 thi-l relA1	Our laboratory
	*L. lactis* NZ9000	Expressing recombinant plasmids	[28]
	*E. coli* DH5α/pMG36e-usp45-*bglA*	*E. coli* DH5α with pMG36e-usp45-*bglA*	This study
	*E. coli* DH5α/pMG36e-usp45-*bglB*	*E. coli* DH5α with pMG36e-usp45-*bglB*	This study
	*E. coli* DH5α/pMG36e-usp45-*bglA*-usp45-*bglB*	*E. coli* DH5α with pMG36e-usp45-*bglA*-usp45-*bglB*	This study
	*L. lactis* NZ9000/pMG36e	*L. lactis* NZ9000 with pMG36e	This study
	*L. lactis* NZ9000/pMG36e-usp45-*bglA*	*L. lactis* NZ9000 with secretory expression BglA	This study
	*L. lactis* NZ9000/pMG36e-usp45-*bglB*	*L. lactis* NZ9000 with secretory expression BglB	This study
	*L. lactis* NZ9000/pMG36e-usp45-*bglA*-usp45-*bglB*	*L. lactis* NZ9000 with secretory expression Bgl	This study
Plasmids	pMG36e	Emr; expression vector with the P32 promoter, multiple cloning sites (MCFs), and prtP translational terminator	[19]
	pMG36e-usp45-*bglA*	Emr; secretory expression of BglA	This study
	pMG36e-usp45-*bglB*	Emr; secretory expression of BglB	This study
	pMG36e-usp45-*bglA*-usp45-*bglB*	Emr; secretory expression of fusion Bgl	This study

**Table 2 microorganisms-11-01387-t002:** Amplification primers used in this study.

Primer	Sequence (5′-3′)	Restriction Enzyme
*bglA*-F	AACTGCAGAGAGCGCAAAAAAAAGATTATCTCAGCTAATGACTATTTTTCAATTTCCGC	*Pst* I
*bglA*-R	TCAAGCTTCTTTGTTTAGCGTC	*Hind* III
*bglB*-F	ATGCTGCAGAATACCTTTATATTTC	*Pst* I
*bglB*-R	CCCAAGCTTGCGCAAAAAAAAGATTATCTCAGCTACTTTTCTATTTAAAACCCG	*Hind* III
*bglA*-R1	AACTGCAGAAGAAGGAGATATACATGCAAAAAAAAGATTATCTCAGCTAATGACTATTTTTCAATTTCCGC	*Pst* I
*bglB*-F1	CCCAAGCTTCCCTTTTCTATTTAAAACCCG	*Hind* III

Single underlines indicate sites of enzyme restriction (cuts); wavy lines indicate the RBS; and double underlines indicate the usp45 signal peptide.

**Table 3 microorganisms-11-01387-t003:** Kinetic parameters of recombinant enzymes.

Strain	*V*max (μmol∙min^−1^mg^−1^)	*K*m (μmol∙L^−1^)	*K*cat (s^−1^)	*K*cat/*K*m (L∙s^−1^∙μmol^−1^)
*L. lactis NZ9000*/pMG36e-usp45-bglA	1343	38.61	434.8	11.26
*L. lactis NZ9000*/pMG36e-usp45-bglB	1254	43.99	348.7	7.93
*L. lactis NZ9000*/pMG36e-usp45-bglA-usp45-bglB	1347	34.8	1016	29.2

## Data Availability

No new data were created or analyzed in this study. Data sharing is not applicable to this article.

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
