# Peer review of "Expression of β-Glucosidases from the Yak Rumen in Lactic Acid Bacteria: A Genetic Engineering Approach"

_microorganisms, 2023, doi:10.3390/microorganisms11061387_

Round 1
Reviewer 1 Report
Dear authors the article is interesting. However in present form need attention and some elements should be improved. The comments are listed below.
1. Line 58-69; The clear aim of study should be added. Author should highlight the main reason of study conduction without not important experimental details. Now it is difficult to understand what is the main reason of study – performing of fusion glucosidase with or lactobacillus strain with this kind of enzyme ?
2. Figure 1. The vectors plots are confusing, especially for fusion Bgl. Could authors explain why additional signal sequence and ribosome binding protein is added before BglB?. I strongly recommend use a professional software for vectors drawing such as SnapGene viewer, Serial cloner, APE etc. Now the plots suggest that vector allows for expression of BglA and BglB simultaneously.
3. The multiple alignment of amino acid sequences with highlighting structural differences between analysis enzymes should be added for better explanation the differences in the determined operational parameters (pH optima, substrate specificity).
4. Does the analyzed enzymes present similar type of folding and catalytic action ?
5. Line; The enzymes units activity definition should be added according to rules of IUPAC.
6. RAC , need explanation.
7. Line 150; with the recombinase supernatant. What authors means “recombinase”
8. The information about protein concentration in post culturing media should be added as the graph or table for showing difference in protein expression level. Level of expression of enzymes should be normalized by total amount of protein in media or precipitates.
9. Figure 3. What authors means “SDS-PAGE of the recombinant strains”. The title should fit to presented results. Moreover the information about amount of protein loaded on each track should be added. Authors could also perform an densitometry analysis with using a GelAnalyzer software for better visualization of difference in band intensity.
10 Why molecular mass of Bgl is 75 kDa? The molecular mass BglA and BglB is 55 kDa and sum of it is 110 kDa ?
11. Figure 4, 5. Authors should add information about substrate used for determination of thermal and pH optima.
12. Line 260-270, The part of discussion should be improved. In present form is very chaotic and difficult to read and understand what author would like to express.
13. Line 295; Authors should clearly state that the hybrid enzyme present better thermal stability compare to…..
14. Enzymes have optimal conditions for its stability and activity that can overlap but not necessary.
Authors preformed only pH catalytic optima.
15. The principal parameters such as Km, Vmax and Kcat should be determined for all analyzed enzymes.
Author Response
Point 1: Line 58-69; The clear aim of study should be added. Author should highlight the main reason of study conduction without not important experimental details. Now it is difficult to understand what is the main reason of study – performing of fusion glucosidase with or lactobacillus strain with this kind of enzyme ?
Response 1: Agreed. It has been modified according to the suggestion. (Line65-67)
Point 2: Figure 1. The vectors plots are confusing, especially for fusion Bgl. Could authors explain why additional signal sequence and ribosome binding protein is added before BglB?. I strongly recommend use a professional software for vectors drawing such as SnapGene viewer, Serial cloner, APE etc. Now the plots suggest that vector allows for expression of BglA and BglB simultaneously.
Response 2: Thanks, Maps of the three recombinants plasmids generated using SnapGene (version5.2.4) in this study are shown in Figure 1abc. (Line180-181)
Point 3: The multiple alignment of amino acid sequences with highlighting structural differences between analysis enzymes should be added for better explanation the differences in the determined operational parameters (pH optima, substrate specificity).
Response 3: Thank you to the reviewer for your comments,Using Blast on NCBI, it was found that the bgla and bglb sequences are 100% identical to Paenibacillus polymyxa (M60210.1 and M60211.1).
Point 4: Does the analyzed enzymes present similar type of folding and catalytic action?
Response 4: Thank, we have not analyzed enzymes present similar type of folding and catalytic action
Point 5: Line; The enzymes units activity definition should be added according to rules of IUPAC.
Response 5: Agreed. The enzymes units activity was consistent with literature [30,31].
Point 6: RAC , need explanation.
Response 6: Thanks, It has been expand
Point 1: Line 150; with the recombinase supernatant. What authors means “recombinase”
Response 7: Yes.
Point 8: The information about protein concentration in post culturing media should be added as the graph or table for showing difference in protein expression level. Level of expression of enzymes should be normalized by total amount of protein in media or precipitates.
Response 8: Agreed. We were determined the concentration of recombinase supernatant using BCA method. The enzymes activity was determined with the same amount of enzymes.
Point 9: Figure 3. What authors means “SDS-PAGE of the recombinant strains”. The title should fit to presented results. Moreover the information about amount of protein loaded on each track should be added. Authors could also perform an densitometry analysis with using a GelAnalyzer software for better visualization of difference in band intensity.
Response 9: Agreed. The band with a molecular mass was analysis using Quantity One 1-D analysis software (version 4.6.2). We were determined the concentration of recombinase supernatant using BCA method
Point 10: Why molecular mass of Bgl is 75 kDa? The molecular mass BglA and BglB is 55 kDa and sum of it is 110 kDa ?
Response 10: Thanks, We also wondered about this part of the results and repeated many experiments, and the results were still true. Mattice W L mainly analyzes SDS-PAGE complex conformation from secondary structure. Comparing the different complex models, a new model is proposed to explain the reasons for the low apparent molecular weight of some alkaline proteins in SDS-PAGE. This literature may indicate the sum of Bgl less than sum of BglA and BglB.
Mattice WL, Riser JM, Clark DS. Conformational properties of the complexes formed by proteins and sodium dodecyl sulfate. Biochemistry. 1976 Sep 21;15(19):4264-72. doi:10.1021/bi00664a020. PMID: 963036.
Point 11: Figure 4, 5. Authors should add information about substrate used for determination of thermal and pH optima.
Response 11: Agreed. It has been modified according to the suggestion. (Line164)
Point 12: Line 260-270, The part of discussion should be improved. In present form is very chaotic and difficult to read and understand what author would like to express.
Response 12: Agreed. It has been modified according to the suggestion.
Point 13: Line 295; Authors should clearly state that the hybrid enzyme present better thermal stability compare to…..
Response 13: Thanks, it It has been modified. (Line330-331)
Point 14: Enzymes have optimal conditions for its stability and activity that can overlap but not necessary.
Response 14: Thanks, Authors preformed only pH catalytic optima.
Point 15: The principal parameters such as Km, Vmax and Kcat should be determined for all analyzed enzymes.
Response 15: Agreed. It has been added according to the suggestion. (line 156-161,252-266)
Reviewer 2 Report
Please check the effect of substrates (Fig. 4) - in text you have 1% salicylic acid in another place 1% salicin, and on the figure 1% saligenin. And this is still the same substance.
Please check also how you write the name of strains: are not always in italic text, once the abbreviation is written, once the full name.
Reviewer 3 Report
The authors have demonstrated the expression of b-glucosidases derived from yak rumen in LAB. Furthermore, authors demonstrated effect of of temperature, pH and metal ions on its activity. Although the topic is interesting since b-glucosidase enzymes are critical for cellulose hydrolysis but there are many caveats that are needed to be addressed prior to publication. Here are comments to improve the manuscript:
a) The manuscript is very poorly written, presentation style must be improved to make it comprehensible for readers. Because of that it is really difficult to confirm hypothesis and conclusion made by the authors.
b) I specifically miss the motivation behind using these two glucosidases. Are these glucosidases are better than the already known and tested ones. I will recommend the authors to include commercial glucosidases or already known glucosidases as positive control to demosntrate what benefits these glucosidases bringing that other's cannot provide.
c) LAB also has native glucosidases. I am surprised that there is no hydrolysis in Fig 2 with just LAB strain. Are these LAB are modified ones with no inherent b-glucosidase activity?
d) The plasmid maps in Fig 1 is not making any sense. These are surely not the true representation of plasmid maps and gene orientation and raising questions on cloning strategy.
e) BglA and bglB sizes are 55 kDa, Fusion protein is just 75 kDa. That does not sound quite right.
f) There are too many grammatical and spelling error.
g) Introduction and discussion is not informative and talk a lot about cloning, expression vector. I would recommend to include details regarding b-glucosidase application instead and towards more commercial aspect instead.
Reviewer 4 Report
Dear Authors
The topic of research paper is interesting and the studies performed are quite satisfying. However, there are few concerns regarding the overall work so, I suggest to improve on all those points as mentioned in comments file attached herewith.

Round 2
Reviewer 1 Report
Dear authors, the manuscript was significantly improved. However some elements still need clarification. The comments are listed below;
1. Could authors explain clearly in manuscript why the RBS site is placed two times in vector of fusion bglA-bglB, ( figure 1 c)? I recommend to add in SnapGene information about start and stop codons especially in case of bgla+bglB vector. The construct DNA sequence and product aminoacid sequence should be also placed in supplementary materials.
2. Figure nr 2. The caption could be changed to; The analysis of recombinant enzymes activity by using agar plates with 0.1 % sodium carboxymethyl cellulose CMC-Na stained Congo red. The halos dimeter is proportional to enzymes activity. To each well was transferred the enzyme volume….. with concentration mg/ml of recombinant enzymes. 1: Bacterial supernatant of L. lactis NZ9000/pMG36e-usp45-bglA; 2: Bacterial supernatant of L. lactis NZ9000/pMG36e; 3: Bacterial supernatant of L. lactis NZ9000/pMG36e-usp45-bglA-usp45-bglB; 4: Bacterial supernatant of L. lactis NZ9000/pMG36e- usp45-bglB; 5: Negative control ddH2O; 6: Bacterial supernatant of L. lactis NZ9000/pMG36e; 7: β- glucosidase standard
3. Line 210-217. The reason of unexpected size of chimeric enzyme is still unclear. Authors should more deeply analysis this findings. The mass of recombinant in case of fusion-protein should be at least 110 kDa ?
Another question is that it was fusion protein, means enzymes subunits linked by linker that consist of a few amino acids. It seems to be that not. Authors wrote (301-303) that addition of RBS allows for generation two ORF ? Or simultaneously expression two independent enzymes in one ORF ?
I strongly recommend to check predicted ORF products by SnapGene that can show why the final protein mass is 75 kDa.
4. Line 243 What authors means; activity peaked at a temperature of 50°C ? It is better to use - the highest activity was observed at …
5. Table 3. Authors could add information about used substrate to parameter determination. The footer below the table. The concentration of substrate was in mol per ml or mol per l?. The table should be in subsection 3.4. Please uniform also units in material and methods now the concentration of sialicin is nM, but per mL or L?
6 .Line 351-353; How the binding efficiency was measured. Authors rather thinking about higher activity by slightly higher affinity (lower Km) how is visible in table 3.
7. Line 368; The Vmax, Km, Kcat, and Kcat/Km of the three recombinant strains were analyzed. The sentence seem to be not complete. Authors should rather give information that this parameters was changed or not comparing the single enzymes and fusion-enzyme.
Author Response
请参阅附件。

Reviewer 3 Report
The authors have provided point to point response to all my comments raised in my previous review to my satisfaction. I have no further comments therefore I endorse manuscript for publication.
Author Response
-
-
Thank you for your review of this article, and best wishes.
-
Reviewer 4 Report
Dear Authors,
I think almost every point has been taken under consideration and improved the manuscript. Now, the revised version can be accepted for publication.
Author Response
-
感谢您对本文的审阅,并致以最良好的祝愿。